# Demonstration of the Systematic Evaluation of an Optical Lattice Clock Using the Drift-Insensitive Self-Comparison Method

**Chihua Zhou** [1,2], **Xiaotong Lu** [1,2], **Benquan Lu** [1], **Yebing Wang** [1] and **Hong Chang** [1,2,*]

1   Key Laboratory of Time and Frequency Primary Standards, National Time Service Center, Chinese Academy of Sciences, Xi'an 710600, China; zch_russ@163.com (C.Z.); 13980960458@163.com (X.L.); lubenquan@126.com (B.L.); wangyebing@ntsc.ac.cn (Y.W.)
2   School of Astronomy and Space Science, University of Chinese Academy of Sciences, Beijing 100049, China
*   Correspondence: changhong@ntsc.ac.cn; Tel.: +86-029-83890852

**Abstract:** The self-comparison method is a powerful tool in the uncertainty evaluation of optical lattice clocks, but any drifts will cause a frequency offset between the two compared clock loops and thus lead to incorrect measurement result. We propose a drift-insensitive self-comparison method to remove this frequency offset by adjusting the clock detection sequence. We also experimentally demonstrate the validity of this method in a one-dimensional $^{87}$Sr optical lattice clock. As the clock laser frequency drift exists, the measured frequency difference between two identical clock loops is $(240 \pm 34)$ mHz using the traditional self-comparison method, while it is $(-15 \pm 16)$ mHz using the drift-insensitive self-comparison method, indicating that this frequency offset is cancelled within current measurement precision. We further use the drift-insensitive self-comparison technique to measure the collisional shift and the second-order Zeeman shift of our clock and the results show that the fractional collisional shift and the second-order Zeeman shift are $4.54(28) \times 10^{-16}$ and $5.06(3) \times 10^{-17}$, respectively.

**Keywords:** optical lattice clocks; strontium atoms; optical lattices; laser cooling and trapping

## 1. Introduction

An optical lattice clock is not only a promising device to generate the second in the future due to its ultra-low uncertainty and instability [1–4], but also a powerful tool to observe physical phenomena such as verifying the general relativity [5–7], testing the Lorentz symmetry [8], detecting gravitational wave [9], and searching the dark matter [10–12]. The space optical lattice clock, which has been proposed by the European Space Agency (ESA) program [13–15], not only carries out geodesy and high-precision measurement of gravitational potential and gravitational redshift [16,17], but also improve the positioning accuracy of the global position system (GPS) and develops deep space navigation [18].

In terms of the uncertainty evaluation of an optical lattice clock, many systematic shifts are measured using the self-comparison method [19,20], such as the lattice AC Stark shift, the collisional shift, the clock laser AC Stark shift and the second-order Zeeman shift [21–25]. However, the drifts of the clock laser frequency and the stray electromagnetic field around the cold-atoms ensemble will lead to the self-comparison measurement error (SCME) [26]. The SCME, generally dominated by the clock laser frequency drift, leads to a frequency offset, of which the magnitude depends on the drift rate and the duration of the clock feedback cycle, and thus, causes incorrect measurement result of the self-comparison method. When the frequency drift rate changes regularly and slowly, this error can be reduced by adding a second-order integral loop to compensate the clock laser frequency using the acoustic optical modulator (AOM) [27], and the residual SCME can be well below $10^{-17}$ [24]. However, when the drift rate varies irregularly or fast, the residual

SCME of this frequency compensation method could prevent the measurement accuracy of the self-comparison below $10^{-17}$. As the optical lattice clock operates outside the laboratory (the transportable optical lattice clocks and even the space optical lattice clocks), the complicated and changeable environment requires us to find more efficient techniques to eliminate the SCME.

In this paper, we propose a drift-insensitive self-comparison (DISC) method to cancel the SCME. By adjusting the interrogation sequence of the self-comparison method, the SCME is cancelled in every clock feedback cycle. The validity of this method is experimentally verified in $^{87}$Sr optical lattice clock where the clock laser frequency drift dominates the drifts. Furthermore, with Rabi spectroscopy, the collisional shift and the second-order Zeeman shift in our system are carefully measured by using the DISC method.

## 2. The Principle of the Drift-Insensitive Self-Comparison Method and the Experimental Setup

### 2.1. The Principle of the Drift-Insensitive Self-Comparison Method

The self-comparison method is that two clock loops ($R_1$ and $R_2$) alternately operate in the time domain. As shown in Figure 1a, a clock feedback cycle of the traditional self-comparison (TSC) method contains four clock detection cycles (the duration of the clock detection cycle is 1 s in this experiment). In the first and the second clock detection cycles, where the clock operates in the systematic parameter of *para*.1, the initial clock laser frequency is set to $f_{01} - \delta/2$ and $f_{01} + \delta/2$, respectively, where the $\delta$ is the full width at half maximum (FWHM) of the spectral peak, and $f_{01}$ corresponds to the center frequency of the spectral peak of the $R_1$. After clock excitations, the excitation fractions of $P_1$ and $P_2$ are obtained, and thus the frequency correction can be calculated by $\Delta f_1 = (P_2 - P_1)\delta/(2P_{max})$, where $P_{max}$ is the maximum excitation fraction. The corrected frequency of $f_{01N} = f_{01} + \Delta f_1$ is closer to the resonance of the $R_1$. In the same way, after the third and fourth clock detection cycles, where the clock operates in the systematic parameter of *para*.2, $f_{02N} = f_{02} + \Delta f_2$ can be obtained. The frequency difference between the $R_1$ and $R_2$ can be expressed by $\Delta v = f_{02N} - f_{01N}$ which is determined every four clock detection cycles. By closed-loop operation, the uncertainty of the $\Delta v$ can be reduced, and thus the influence of a certain parameter on the clock transition frequency can be measured with high precision.

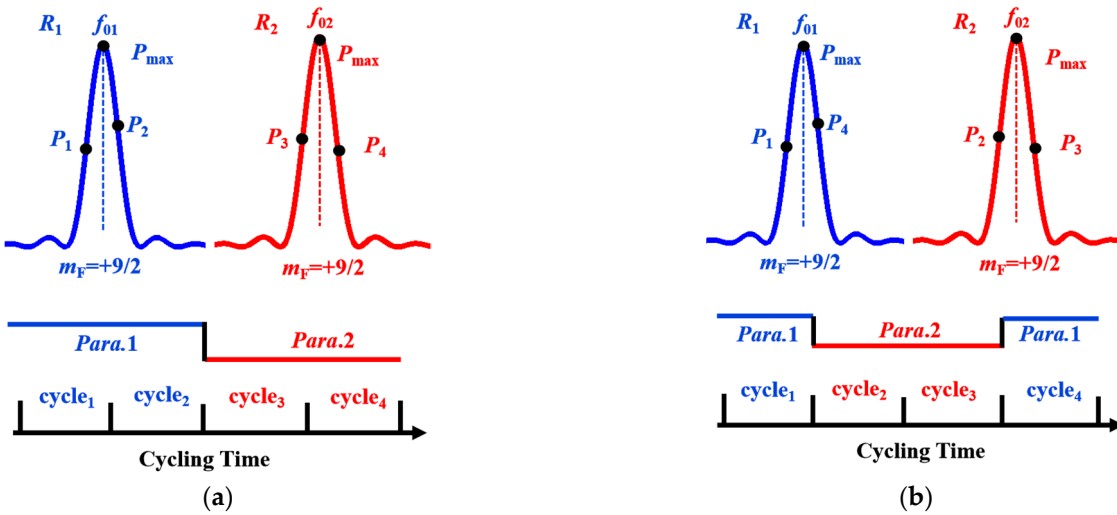

**Figure 1.** Schematic diagram of self-comparison method. (**a**) Traditional self-comparison method. (**b**) Drift-insensitive self-comparison method. *Para*.1 and *Para*.2 represent two different values of the same system parameter (such as atomic density, magnetic field strength); $R_1$ and $R_2$ indicate the two clock loops that operate in an interleaved way in the time domain; $f_{01-02}$ correspond to the central frequency of the spectral peaks of $R_1$ and $R_2$, respectively; $P_{max}$ indicates the maximum excitation fraction; and $P_{1-4}$ are the clock transition probabilities of the first to forth clock detection cycles, respectively.

Obviously, in the process of TSC method, because the $R_1$ and $R_2$ successively operate in the time domain and the amount of the frequency drift accumulates over time, the frequency correction, caused by the frequency drift, of $R_2$ is twice that of $R_1$. To take the measurement of the collisional shift as an example, in the first two clock detection cycles, the atomic density is higher (*H*), while in the second two clock detection cycles, the atomic density is lower (*L*). If the drift rate is $\zeta$ Hz/s and the measurement sequence is *HHLL*, the drift-induced frequency offset for $R_1$ is about $\Delta_{s1} = (\zeta + 2\zeta)/2 = 3\zeta/2$, and in terms of $R_2$, the drift-induced frequency offset is about $\Delta_{s2} = (4\zeta + 3\zeta)/2 = 7\zeta/2$. Therefore, the SCME is about $\Delta_s = \Delta_{s2} - \Delta_{s1} = 2\zeta$ as the density shift is measured by the TSC method.

According to the previous analysis, a simple but high-efficiency way (DISC method), which is realized by changing the detection sequence as *HLLH* (or *LHHL*) (as shown in Figure 1b), can be used to cancel the SCME and achieve drift-insensitive self-comparison measurement. When the detection sequence is *HLLH*, the drift-induced frequency offset is $\Delta'_{s1} = (\zeta + 4\zeta)/2 = 5\zeta/2$ for $R_1$ and $\Delta'_{s2} = (2\zeta + 3\zeta)/2 = 5\zeta/2$ for $R_2$. Thus, the SCME is $\Delta'_{s1} = \Delta'_{s1} - \Delta'_{s2} = 0$, indicating that the SCME is cancelled. The DISC method will not complicate the existing devices or consume extra time. Meanwhile, in terms of the cancellation of the SCME, the DISC method, which updates the frequency drift rate every four clock detection cycles, is expected to work better than the traditional way described in reference [27], which typically needs forty clock detection cycles to update the frequency drift rate. It's worth noting that the DISC method cannot be used when the drift rate is very high (for example, the total frequency drift is larger than the FWHM of the spectrum in one clock feedback cycle) due to bad locking.

### 2.2. Description of the Experimental Setup of $^{87}Sr$ Optical Lattice Optical Clock

We experimentally verify the validity of the DISC method based on the one-dimensional $^{87}Sr$ optical lattice clock, the details of which is described in reference [26,28]. After two-stage laser cooling, about $8 \times 10^4$ atoms are loaded into a horizontal one-dimensional optical lattice. Using the Pound–Drever–Hall (PDH) technique, the lattice laser wavelength is stabilized to an ultra-low expansion (ULE) cavity at 813.42 nm where the atomic polarizability of the clock ground state and the excited state is same, the so called "magic wavelength" [29]. Following that, the atoms are spin-polarized to the Zeeman sublevels of $\left|^1S_0, m_F = +9/2\right\rangle$ with a spin-polarized purity of more than 99%. The clock laser is locked to an ULE cavity with a finesse of 400,000 by PDH stabilization at 698.44 nm corresponding to the $5s^2\,^1S_0 \rightarrow 5s5p\,^3P_0$ transition. The line-width of the clock laser is about 1 Hz obtained by beating with another similar clock laser system. The polarization of the clock laser and the lattice laser is linear and the direction is along the one of the magnetic field quantization axis which is parallel to the gravity. According to the resolved sideband spectroscopy, the longitudinal and radial temperatures of atoms trapped in the lattice are 2.9 and 2.7 μK, respectively. Additionally, the misalignment angle between the clock laser beam and the lattice light is about 6 mrad extracted from the Rabi oscillation of the carrier transition [30,31].

## 3. Experimental Results

### 3.1. Self-Comparison Measurement Error Cancellation using the Drift-Insensitive Self-Comparison Method

The SCME is evaluated by measuring the frequency difference between the two identical clock loops (the systematic parameters of $R_1$ and $R_2$ are the same) with the TSC and DISC methods, respectively. If there are no drifts at all, the expected frequency difference between $R_1$ and $R_2$ is zero. Figure 2a shows the frequency difference between $R_1$ and $R_2$ by TSC and DISC methods, respectively, without changing any experimental parameters but the detection order. The SCME of the TSC method is $(240 \pm 34)$ mHz, while the SCME measured using the DISC method is $(-15 \pm 16)$ mHz, where the measurement uncertainty is given by the last point of their respective total Allan deviation of the self-comparison instabilities. With the DISC method, the SCME is consistent with zero, indicating that the

DISC method almost completely removes the frequency offset caused by the drifts even if the clock laser frequency drift is nonlinear and dramatically change as shown in the inset of Figure 2a. Furthermore, the comparison data shown in Figure 2a also demonstrate that the magnitude of the frequency difference fluctuation with the DISC method is significantly smaller than the TSC method. Therefore, the self-comparison instability of the DISC method is lower than the one using the TSC method. As shown in Figure 2b, the self-comparison instability of the TSC method is $5.6 \times 10^{-15} \tau^{-0.5}$ ($\tau$ is the averaging time), while the self-comparison instability of the DISC method is $3.1 \times 10^{-15} \tau^{-0.5}$, indicating that the DISC method cannot only cancel the SCME, but also improve the measurement accuracy when the averaging time exceeds 100 s. In order to eliminate the frequency offset caused by the drifts, we also try to use the AOM to compensate the frequency drift of the clock laser in the TSC method, where the frequency sweeping rate is calculated every 40 s. However, the frequency compensation method cannot completely remove the frequency offset and the residual frequency offset is more than 30 mHz which is caused by the rapidly changed drift rate in our system.

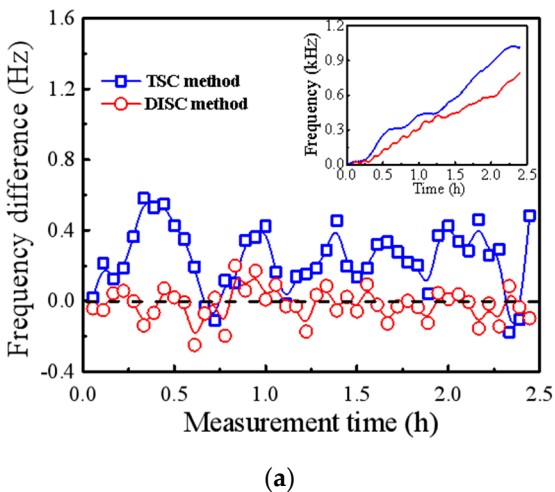

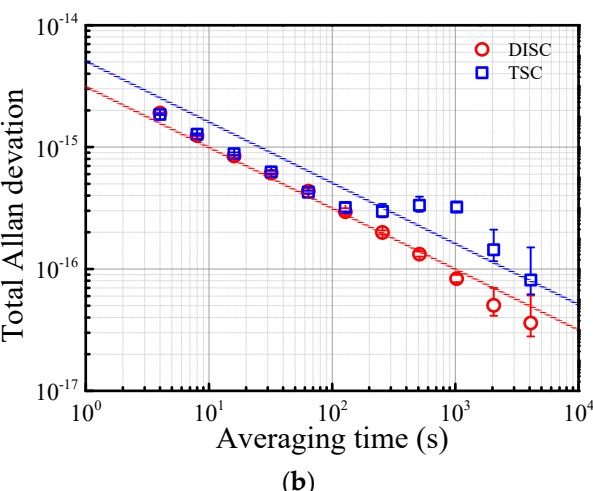

(**a**)          (**b**)

**Figure 2.** (**a**) Comparison of the traditional self-comparison and the drift-insensitive self-comparison methods. Red circles are measured using the drift-insensitive self-comparison method, the blue squares are obtained using the traditional self-comparison method, and the black dashed line indicate zero frequency difference. Each experimental data point is the average value over 200 s measurement time, and the inset shows the clock laser frequency corrections over the self-comparison process (the blue and red lines correspond the traditional self-comparison and the drift-insensitive self-comparison methods, respectively). (**b**) Comparison of self-comparison instabilities. The blue squares indicate the instability of the traditional self-comparison method and the red circles represent the instability of the drift-insensitive self-comparison method. The error bars correspond to $1\sigma$ standard deviation and the solid lines are the linear fitting with a fixed slope of $-0.5$. The instability falls again above $10^3$ s due to the approximately linear drift in the whole process.

### 3.2. The Collisional Shift Evaluation

With Rabi spectrum, we further measure the collisional shift using the DISC method in our $^{87}$Sr clock. The clock transition benefits from the detection of thousands of trapped cold-atoms simultaneous optical transitions, resulting in optical lattice clocks with ultra-low quantum projection noise limit. On the other hand, for a one-dimensional optical lattice clock, large number of atoms in the same lattice site will cause the collisional shift. Even for Fermions, the collisional shift can be more than $1 \times 10^{-16}$ [32,33], indicating that the collisional shift should be carefully evaluated. As the trap potential and atomic temperature keep unchanged, the collisional shift depends on the atomic density and the excitation fraction ($P_e$). The way of measuring this shift is shown in Figure 1b, where we set *Para*.1 as high density ($I_1$) and *Para*.2 as low density ($I_2$) and the atomic density is changed by varying the current of the Zeeman slower, which changes the loaded atoms of the first-stage cooling and eventually changes the total atomic number of the lattice. $I_1$ and $I_2$,

which are in direct proportion to the atomic density, represent the fluorescence intensity collected by the photomultiplier (PMT). In order to avoid the influence of the variation of the total atomic number, for the *i*th clock feedback cycle, the comparison result is divided by $\Delta I_i = I_{1i} - I_{2i}$, to obtain the collisional shift of unit fluorescence intensity (marked by $\Delta_{uf}$) [20,24], where the $I_{1i}$ and $I_{2i}$ correspond to the fluorescence intensity under the conditions of *Para*.1 and *Para*.2, respectively. Additionally, in terms of the regular operation of the clock, the collisional shift is calculated by multiplying the $\Delta_{uf}$ by the intensity $I_{nor}$ that corresponds to the fluorescence intensity as the clock regularly operates. The gain of the PMT and the detection laser intensity remain constant during the whole measurement process.

With Rabi spectrum, the collisional shifts under different excitation fraction are measured as shown in Figure 3a. Typically, the atoms are prepared in $^1S_0$ state (the ground state), but those points that the excitation fractions are larger than 0.4 are realized by preparing the atoms in $^3P_0$ state (the excited state). By linear fitting [34,35], the relationship of $\Delta_{uf}$ and $P_e$ is determined as $\Delta_{uf} = 18.5(18)P_e - 11.6(7)$. Thus, for the regular clock operation of which the expected collisional shift is about $45\Delta_{uf}$, 1% change in the excitation fraction will lead to a change in fractional collisional shift of $2 \times 10^{-17}$.

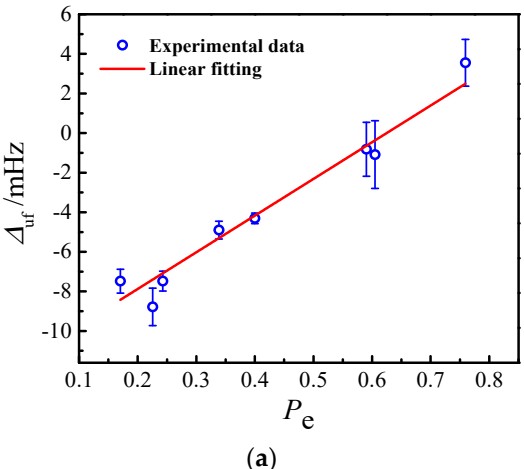

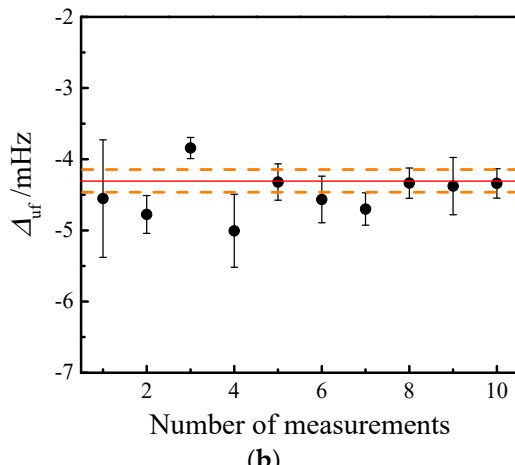

(**a**)　　　　　　　　　　　　　　　　　　　　(**b**)

**Figure 3.** (**a**) The collisional shift of unit fluorescence intensity ($\Delta_{uf}$) at different excitation fraction. The error bars are given by the last point of the self-comparison instability. (**b**) The measurements of $\Delta_{uf}$ when the excitation fraction is about 0.4. The measurement uncertainty is also given by the last point of the self-comparison instability. The solid line indicates the weighted average of the ten measurements of which the value is same as the corresponding data in (**a**) where $P_e$ is about 0.4. The dashed lines represent the $1\sigma$ standard deviation of the average value that has been multiplied by the square root of the reduced-chi-square $\chi^2_{red} = 2.06$.

As the clock regularly operates, the average excitation fraction $P_{e\text{-}nor}$ is about 0.4 which corresponds to the point of half the maximum excitation. Thus, we carefully evaluate the collisional shift when the excitation fraction is $P_{e\text{-}nor}$ as shown in Figure 3b. The weighted mean of the ten measurements is 4.31(27) mHz per unit fluorescence intensity, where the measurement uncertainty has been multiplied by the square root of the reduced-chi-square $\chi^2_{red} = 2.06$. Thus, as the clock regularly runs, the corresponding fractional collisional shift is $4.54 \times 10^{-16}$ with an uncertainty of $2.8 \times 10^{-17}$. The residual SCME, caused by fast changed drift, will lead random frequency offset to each measurement, which eventually increases the uncertainty of the measurements. However, the experimental result shows that with the DISC method, the collisional shift is obtained with a precision of $2.8 \times 10^{-17}$, though we have not compensated the frequency drift.

### 3.3. The Second-Order Zeeman Shift Evaluation

The DISC method can also be applied in the closed-loop operation of optical lattice clocks. For removing the first-order Zeeman shift, the lattice light vector shift and line

pulling shift, the clock laser frequency of the $^{87}$Sr optical lattice clock is usually stabilized to the average frequency of the transitions of $m_F = +9/2 \rightarrow m_F = +9/2$ and $m_F = -9/2 \rightarrow m_F = -9/2$. The magnitude of the bias magnetic field, which defines the magnetic field quantization axis, can be accurately extracted from the frequency gap between the two transitions. Benefiting from the DISC method, this frequency gap can be precisely extracted without the frequency offset caused by the clock laser frequency drift. Figure 4 shows the frequency gap of the $m_F = +9/2$ and $m_F = -9/2$ during the closed-loop operation of our clock with the DISC operation. Herein, *Para*.1 and *Para*.2 correspond to the $m_F = +9/2 \rightarrow m_F = +9/2$ and $m_F = -9/2 \rightarrow m_F = -9/2$ transitions, respectively, and the clock detection cycle is 0.6 s (the duration of the clock laser is 0.15 s). The average frequency gap is 297.8(8) Hz, where the uncertainty indicates the 95% confidence interval of the mean, and the corresponding magnetic field intensity is 305.2(8) mG [36]. As the second-order Zeeman coefficient is $-23.37(3)$ MHz/T$^2$ [37], the fractional second-order Zeeman shift is $5.06(3) \times 10^{-17}$.

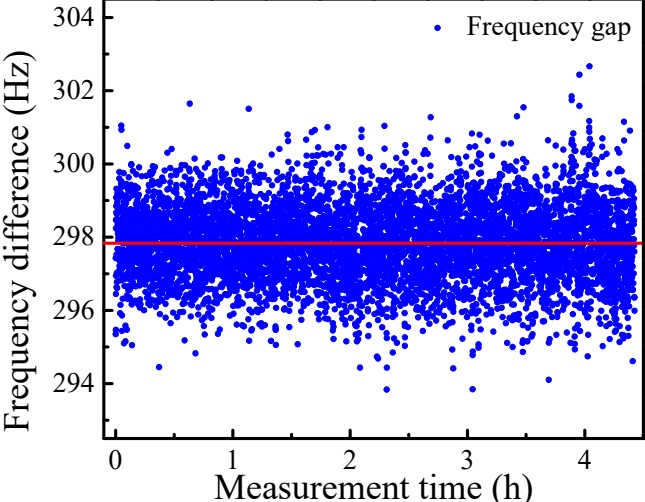

**Figure 4.** The frequency gap between $m_F = +9/2$ and $m_F = -9/2$ during closed-loop operation of the clock using the DISC method. The red solid line indicates the average gap by linearly fitting the experimental data. The magnetic field drift rate is $-4(6) \times 10^{-6}$ mG/s extracted from the experimental data, where the uncertainty indicates the 95% confidence interval and the magnetic field drift rate agree with zero.

## 4. Conclusions

In summary, we propose and experimentally demonstrate a drift-insensitive self-comparison method to eliminate the self-comparison measurement error and evaluate the systematic shifts. By measuring the frequency difference between two identical clock loops, the self-comparison measurement error is $(240 \pm 34)$ mHz using the traditional method, while it is $(-15 \pm 16)$ mHz using the drift-insensitive method. Based on the drift-insensitive self-comparison technique, we use Rabi spectrum to evaluate the collisional shift of our clock. With the DISC method, the collisional shift and the second-order Zeeman shift are evaluated as $4.54(28) \times 10^{-16}$ and $5.06(3) \times 10^{-17}$, respectively. The lattice light and the clock laser AC Stark shifts can also be measured by this method. The DISC method could be widely used in the evaluation of the space clocks that have less possibility to operate the clock under a condition of no drift or a linear drift due to complicated space environment. Combined with the drift compensation process, the SCME can be further suppressed even in a harsh experimental environment.

**Author Contributions:** Conceptualization, H.C.; data curation, B.L. and C.Z.; funding acquisition, H.C.; investigation, Y.W.; methodology, C.Z.; project administration, H.C.; resources, H.C.; software, C.Z.; supervision, Y.W.; validation, X.L.; writing—original draft, X.L. and C.Z.; writing—review and editing, B.L. and Y.W. All authors have read and agreed to the published version of the manuscript.

**Funding:** This research is supported by the National Natural Science Foundation of China (61775220), the National Key R&D Program of China (Grant No. 2016YFF0200201), the Key Research Project of Frontier Science of the Chinese Academy of Sciences (Grant No. QYZDB-SSW-JSC004) and the Strategic Priority Research Program of the Chinese Academy of Sciences (Grant No. XDB21030100).

**Institutional Review Board Statement:** Not applicable.

**Informed Consent Statement:** Not applicable.

**Data Availability Statement:** The data presented in this study are available on request from the corresponding author. The data are not publicly available due to department requirements.

**Conflicts of Interest:** The authors declare no conflict of interest.

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
