# Peer review of "Demonstration of the Systematic Evaluation of an Optical Lattice Clock Using the Drift-Insensitive Self-Comparison Method"

_applsci, doi:10.3390/app11031206_

Round 1

Reviewer 1 Report

According to my previous comments, authors have addressed all the issues regarding the description of the methods used and the presentation of the obtained results.

Also, English language and style have been both extensively edited.

My opinion about the paper is that it can be now accepted for publication in the present form.

Author Response

We are grateful for your efforts on our manuscript. 

Reviewer 2 Report

I read the newly revised manuscript by Zhou et al. I find the present version suitable for publication in Applied Sciences. I still have few minor language points which I list below:

(119) If there are no drifts at all, the expected frequency difference...
(120) Figure 2a shows...
(160) ... I1 and I2, which are in direct proportion...
(171) larger than 0.4 are realized...
(199) Benefiting from the DISC method...

I can recommend the publication of the paper in Applied Sciences.

Author Response

(160) ... I1 and I2, which are in direct proportion...

Response:

We replaced “I1 and I2, which is in direct proportion to the atomic density…” by “I1 and I2, which are in direct proportion to the atomic density…”.

(171) larger than 0.4 are realized...

Response:

We replaced “…but those points that the excitation fractions are larger than 0.4 is realized by…” by “…but those points that the excitation fractions are larger than 0.4 are realized by…”.

(199) Benefiting from the DISC method...

Response:

We replaced “Benefiting of the DISC method, this frequency gap…” by “Benefiting from the DISC method, this frequency gap…”.

Thanks for the helpful advices from the reviewer#2. We are grateful for your consideration and look forward to further communication.

This manuscript is a resubmission of an earlier submission. The following is a list of the peer review reports and author responses from that submission.

Round 1

Reviewer 1 Report

The article concerns a problem with the drift of the ultrastable laser that is not rare in the optical clock community and it is the next publication of the group about the removing frequency drifts of the clock laser. But the presented method is not competely new in the clock community moreover the group already presented simulations for removing the drift [ref 25] and there is already other method for elimination such drift as you write in your manuscript [ref 26]. The second part is devoted on the evaluation of the density shift in the optical clock. These results are relevant to the evaluation of the accuracy budget of the clock.

But I found points that the article can be much better:
general: we propose; proposal is a noun
line 26: I think that the optical transitions in neutral atoms are the candidates to redefine the SI second. Optical lattice clock is just a tool.
section2:
This method is not completely new. I suggest to add simulations that confirm this method and write about possible limitations of this method.
line 52: it should be two clocks loops R1 and R2
fig 1b. it is a bit unfortunate. In the first view I see that you change parameter1 to parameter2 during the Rabi spectroscopy. The figure must be more clear.
line91: wrong citation.
line95: misspelling: state instead stats
line102: radial instead radical

section 3.2.
It is described unclear. I highly recommend to rewrite this section in the clear way and with more details:
line138: collisional shift, collision is a noun
line145: Density shift depends on density of atoms and probability excitation as you marked. But you focus only on the probability exctitation dependence. How do you change the atomic density? Did you perform measurements for various atomic densities? It will be good to include also the dependance of the clock frequency on the total atom number in the lattice and calculate the density shift based on the theoretical model presented in the ref 19 and 20. There is no comparison of your results with results from other groups.
line167: it should be: fractional collisional shift
Fig.3. (a): I wonder why there is no point for 0.4 P_e (there is only one point at 0.35 P_e). I suggest to show more points around the excitation fraction when the clock daily runs.
Fig.3. (b): I wonder that these measurements are performed for the same ΔI_i or for various? There is no information about this.
line171-174: Sentences: Thus, the DISC method could be widely used in the evaluation of
the space clocks that have less possible to operate the clock under no drifts or a linear drift due to complicated space environment. Combining with the drift compensation process, the SCME can be further suppressed even in terrible experimental environments.
They should be in the section with conclusions.

I recommend also to change the title of the article. More relevant in this article is the evaluation of the density shift.

Author Response

First of all, we would like to thank the reviewer1 for the helpful suggestions and comments. The responses to the comments are listed as following:

(1) general: we propose; proposal is a noun.

Response: We replaced “proposal” by “propose” in lines 14, 69 and 381.

(2) line 26: I think that the optical transitions in neutral atoms are the candidates to redefine the SI second. Optical lattice clock is just a tool.

Response: Thank you for your thoughtful comments. In the revised manuscript, we replaced “Optical lattice clocks are not only one of the candidates to redefine the second in the future…. and searching the dark matter[9-11]” by “Optical lattice clock is not only a promising device to generate the second in the future due to their ultra-low uncertainty and instability [1-4], but also a powerful tool to observe physical phenomena such as verifying the general relativity [5,6]…and searching the dark matter[9-11]”.

(3) section2: 

This method is not completely new. I suggest to add simulations that confirm this method and write about possible limitations of this method.

Response: We have experimentally demonstrated the validity of the DISC way by comparing with the TSC method. Unlike our previous way that removes the SCME by data processing (in reference [25]), in terms of the SCME cancellation, the DISC method is far more compelling. However, if the drift rate is much large (such as larger than the FWHM of the spectrum during on feedback cycle), this method will be failed due to bad locking. Thus, we added this sentence “It's worth noting that the DISC method cannot be used when the drift rate is very high (for example, the total frequency drift is larger than the FWHM of the spectrum in one clock feedback cycle) due to bad locking” in line 108 to discuss the possible limitations of the DISC method.

(4) line 52: it should be two clocks loops R1 and R2 fig 1b. it is a bit unfortunate. In the first view I see that you change parameter1 to parameter2 during the Rabi spectroscopy. The figure must be clearer.

Response: We replaced the Fig. 1(a) and (b) as following:

(a)                             (b)

Figure 1. Schematic diagram of self-comparison method. (a) Traditional self-comparison method. (b) Drift insensitive self-comparison method. Para.1 and Para.2 represent two different values of the same system parameter (such as atomic density, magnetic field strength), R1 and R2 indicate the two clock loops that interleavedly operate in the time domain, f01-02 correspond to the central frequency of the spectral peaks of R1 and R2 respectively, Pmax indicates the maximum excitation fraction, and P1-4 are the clock transition probabilities of the first to forth clock detection cycles, respectively.

(5) line91: wrong citation.

Response: Thank you for your correction. In the revised manuscript, we replaced “…the details of which is described in reference [22,27]” by “the details of which is described in reference [25,27]” in line 177.

(6) line95: misspelling: state instead stats.

Response: We replaced “stats” by “state”.

(7) line102: radial instead radical.

Response: We replaced “radical” by “radial”.

(8) section 3.2. 

It is described unclear. I highly recommend to rewrite this section in the clear way and with more details: 

line138: collisional shift, collision is a noun.

Response: We replaced “collision shift” by “collisional shift” in the whole revised manuscript.

(9) line145: Density shift depends on density of atoms and probability excitation as you marked. But you focus only on the probability excitation dependence. How do you change the atomic density? Did you perform measurements for various atomic densities? It will be good to include also the dependance of the clock frequency on the total atom number in the lattice and calculate the density shift based on the theoretical model presented in the ref 19 and 20. There is no comparison of your results with results from other groups.

Response: It is hard to directly theoretically calculate the density shift. In term of the uncertainty evaluation of the collisional shift in optical lattice clocks, to experimentally measure this shift is enough. However, it unnecessary to compare the measurement results of the collisional shift between different groups, as apart from atomic density and excitation fraction, this shift also depends on atomic temperature, lattice structure, the misalignment angle between the lattice laser and the clock laser and so on. These factors are generally different between groups, and thus, to compare our results with results from other groups seems meaningless. However, in terms of our clock, these parameters (such as the temperature, the misalignment angle, lattice structure and so on) are same between measurements, indicating that to evaluating the relationships between the collisional shift and the atomic density and the excitation fraction is enough.

       In this experiment, we don’t measure the atomic density directly, where we just measure the density shift per unit fluorescence intensity collected by the photomultiplier (PMT), which is enough for the collisional shift evaluation of an optical lattice clock. To precisely determine the atomic density usually motives by other reasons such as extracting the magnitude of the scattering length of s-wave or p-wave. To reduce the influence of the atomic number drift, we just measured the collisional shift of unit fluorescence intensity, and thus to evaluate the dependance of the clock frequency on the total atom number in the lattice is impossible unless at the cost of decreasing the measurement accuracy.

       In the revised manuscript, we also added “the atomic density is changed by varying the current of the Zeeman slower” to briefly describe how the atomic density is changed.

(10) line167: it should be: fractional collisional shift.

Response: We replaced “fraction collision shift” by “fractional collisional shift”.

(11) Fig.3. (a): I wonder why there is no point for 0.4 P_e (there is only one point at 0.35 P_e). I suggest to show more points around the excitation fraction when the clock daily runs.

Response: When we obtained the data in Fig. 3(a), we did not measure the collisional shift at the excitation fraction corresponding to the regular operation of the clock (Pe=0.4), considering that we will carefully measure it in the following work. The data of Fig. 3(b) can be directly applied in Fig. 3(a). Thus, we just use the weighted average of Fig. 3(b) to supplement the correspond data in Fig. 3(a) in the revised manuscript. The collisional shift per unit fluorescence intensity cannot be extracted as there is no modulation of atomic density when the clock daily runs.

(12) Fig.3. (b): I wonder that these measurements are performed for the same ΔI_i or for various? There is no information about this.

Response: The value of ΔIi, recorded every feedback cycle (4 s), maybe change in each feedback cycle due to the fluctuation of the total atomic number. The value of Δuf is, automatically, obtained by dividing the comparison result (Δfc-i that is obtained every feedback cycle) by ΔIi. We had described it by “…for the ith clock feedback cycle, the comparison result is divided by ΔIi=I1i-I2i, to obtain the collisional shift of unit fluorescence intensity (marked by Δuf)”.

(13) line171-174: Sentences: Thus, the DISC method could be widely used in the evaluation of the space clocks that have less possible to operate the clock under no drifts or a linear drift due to complicated space environment. Combining with the drift compensation process, the SCME can be further suppressed even in terrible experimental environments. They should be in the section with conclusions.’

Response: In the revised manuscript, we removed this sentence to the section of conclusions.

(14) I recommend also to change the title of the article. More relevant in this article is the evaluation of the density shift.

Response: The title was changed to “An evaluation of the collisional shift by the drift insensitive self-comparison method in optical lattice clocks”

We carefully checked the manuscript to avoid any grammar or spelling mistakes. All revisions have been highlighted by using the “Track Changes” function in the revised manuscript.

Reviewer 2 Report

In this work, Zhou et al. introduce a very simple but smart solution to reduce the self-comparison measurement error (SCME) in optical lattice clocks. The solution is based on a different algorithm used for the clock detection sequence, which allows SCME to be significantly reduced with respect to the more commonly used TSC (Tradtional Self-Comparison) technique. Moreover, they demonstrate that their solution allows for a more precise evaluation of the collision shift, which is one of the factors affecting the clock overall accuracy.

I think that this work deserves publication in Applied Sciences journal. The manuscript is well-organized. Methodologies are accurately described. Conclusions drawn are well supported by data, because authors provide reliable results obtained with experiments on a real 87Sr optical lattice clock. A very weak point is English, which has necessarily to be improved; indeed, there are so many grammar mistakes that it’s sometimes impossible to understand the text (e.g. “systemic” instead of “systematic”, “we proposal” instead of  “we propose”, singular verbs used instead of plural ones, and so on). Please carefully check English before re-submission.

By the way, I suggest to publish the paper after minor revision, provided that authors amend some minor issues herewith enlisted:

  • Line 21. Replace here (and in other parts of the manuscript) “systemic” with “systematic”.
  • Line 38. Optical lattice clocks are a very interest topic, but a niche topic as well. I suggest to dedicate few lines here on the description of SCME, the reduction of which is the main target of the authors’ research.
  • Lines 39 and 42. The terms “slowly” and “fast” for the frequency drift rate should be quantified. How slowly can change the drift rate to allow AOM-based compensation to be effective, for example? Authors should elaborate a little bit more on this.
  • Line 54. Symbols that are not subsequently used into expressions or equations should be removed (e.g., TC at line 54, λL at line 94, λP at line 98), leaving indicated only the numeric values.
  • Lines 57-58. When defining FWHM and center frequency, I would refer to “the spectral peak” rather than “the spectrum”.
  • 1 should be modified an integrated with further info. For sake of clarity, I would add the numbers of the detection cycles on the cycling time axis (e.g. “ cycle 1” , “cycle 2”, etc.). I would add the caption “spectral peaks” near the peaks related to the two clocks. Also, add to the figure the symbols R1 and R2 used to indicate the two clocks, and indicate PMax on the top of the peaks.
  • Line 82. To better appreciate the advantage with respect to the 40 detection cycles of TSC, authors should indicate how many detection cycles DISC method is expected to take to update the frequency drift rate.
  • Line 93. Replace “frequency” with “wavelength”.
  • Line 97. It’s not “fineness”, but “finesse”. Please correct.
  • Line 107. Replace “measured” with “evaluated”.
  • Line 112. Authors mention “error bars” of Figure 2(a), but I cannot see any error bar in Figure 2(a). Could authors clarify this point?
  • Line 122. Authors should add “when the averaging time exceeds 100 s”. Indeed, if averaging time is lower than 100 s, instabilities of TSC and DISC are the same, as can be seen from Fig. 2(b).
  • Line 167. Replace “SECM” with “SCME”.
  • Line 172. Replace “possible” with “possibility”.
  • Line 174. I find “terrible” a little bit inappropriate term. Use “harsh”.
  • 3(a). I cannot see in the plot the experimental data referring to Pe = 0.4, which authors introduced (line 162) as the excitation fraction corresponding to the regular operation of the clock. Why? Being Δuf values @ Pe = 0.4 reported in Fig. 3(b) the corresponding average Δuf value should be also reported in Fig. 3(a) for completeness.
  • Conclusion is not a conclusion, but a second abstract. It should be removed in the present form, or replaced with a few considerations different from a mere repetition of the aim of the work and the obtained results. For instance, authors should add the future perspective of their research activity, if any (e.g. are there feasible ways to reduce offsets when drift rate changes rapidly?)

Author Response

First of all, we appreciate the reviewer2 for the helpful suggestions and comments. The responses to the comments are listed as following:

(1) Line 21. Replace here (and in other parts of the manuscript) “systemic” with “systematic”.

Response: We replaced “systemic” by “systematic” in the whole revised manuscript.

(2) Line 38. Optical lattice clocks are a very interest topic, but a niche topic as well. I suggest to dedicate few lines here on the description of SCME, the reduction of which is the main target of the authors’ research.

Response: In the revised manuscript, we added “The SCME, generally dominated by the clock laser frequency drift, leads to a frequency offset, of which the magnitude depends on the drift rate and the duration of the clock feedback cycle, and thus, causes incorrect measurement result of the self-comparison method” in line 39 to briefly describe the SCME.

(3) Lines 39 and 42. The terms “slowly” and “fast” for the frequency drift rate should be quantified. How slowly can change the drift rate to allow AOM-based compensation to be effective, for example? Authors should elaborate a little bit more on this.

Response: In fact, when the frequency drift rate varies with time, any already existed methods cannot completely cancel the SCME. When the frequency drift rate varies fast, the residual SCME will increase to an unacceptable level (when the frequency drift rate changes slowly enough, the residual SCME will not limit the measurement accuracy). Thus, it is unnecessary (and difficult) to quantificationally determine how slowly can change the drift rate to allow AOM-based compensation to be effective, as the drift rate limit the measurement precise. We are so sorry that our previous expressions fail to explain this clearly. Thus, in the revised manuscript, we replaced “When the frequency drift rate changes slowly, this error can be eliminated by adding a second-order integral loop to compensate the clock laser frequency by the acoustic optical modulator (AOM) [26]. However, when the drift rate varies fast, this frequency compensation method cannot completely eliminate the SCME” by “When the frequency drift rate changes regularly and slowly, this error can be reduced by adding a second-order integral loop to compensate the clock laser frequency by the acoustic optical modulator (AOM) [26], and the residual SCME can be well below 10-17 [23]. However, when the drift rate varies irregularly or fast, the residual SCME of this frequency compensation method could prevent the measurement accuracy of the self-comparison below 10-17”.

(4) Line 54. Symbols that are not subsequently used into expressions or equations should be removed (e.g., Tat line 54, λL at line 94, λP at line 98), leaving indicated only the numeric values.

Response: In the revised manuscript, we deleted “TC ”, “λL” and “λP”.

(5) Lines 57-58. When defining FWHM and center frequency, I would refer to “the spectral peak” rather than “the spectrum”.

Response: We replaced “the spectrum” by “the spectral peak” in lines 82-83.

(6) 1 should be modified an integrated with further info. For sake of clarity, I would add the numbers of the detection cycles on the cycling time axis (e.g. “ cycle 1” , “cycle 2”, etc.). I would add the caption “spectral peaks” near the peaks related to the two clocks. Also, add to the figure the symbols R1 and Rused to indicate the two clocks, and indicate PMax on the top of the peaks.

Response: We updated the Fig. 1(a)~(b) as following:

(a)                             (b)

Figure 1. Schematic diagram of self-comparison method. (a) Traditional self-comparison method. (b) Drift insensitive self-comparison method. Para.1 and Para.2 represent two different values of the same system parameter (such as atomic density, magnetic field strength), R1 and R2 indicate the two clock loops that interleavedly operate in the time domain, f01-02 correspond to the central frequency of the spectral peaks of R1 and R2 respectively, Pmax indicates the maximum excitation fraction, and P1-4 are the clock transition probabilities of the first to forth clock detection cycles, respectively.

(7) Line 82. To better appreciate the advantage with respect to the 40 detection cycles of TSC, authors should indicate how many detection cycles DISC method is expected to take to update the frequency drift rate.

Response: We replaced “…, the DISC method is expected to work better than the traditional way described in reference [19], which typically needs forty clock detection cycles to update the frequency drift rate” by “…, the DISC method, which updates the frequency drift rate every four clock detection cycles, is expected to work better than the traditional way described in reference [19], which typically needs forty clock detection cycles to update the frequency drift rate”.

(8) Line 93. Replace “frequency” with “wavelength”.

Response: We replaced “frequency” by “wavelength” in line 184.

(9) Line 97. It’s not “fineness”, but “finesse”. Please correct.

Response: We replaced “fineness” by “finesse” in line 188.

(10) Line 107. Replace “measured” with “evaluated”.

Response: We replaced “measured” by “evaluated” in line 198.

(11) Line 112. Authors mention “error bars” of Figure 2(a), but I cannot see any error bar in Figure 2(a). Could authors clarify this point?

Response: The error bars correspond to the showed frequency offsets of (240±34) mHz for the TSC method and (-15±16) mHz for the DISC method. To make it clearer, we replaced “The SCME of the TSC method is (240±34) mHz, while the SCME measured by the DISC method is (-15±16) mHz. Herein, the error bars are given by the last points of their respective total Allan deviation of the self-comparison instabilities.” by “The SCME of the TSC method is (240±34) mHz, while the SCME measured by the DISC method is (-15±16) mHz, where the measurement uncertainty indicates the 1σ standard of the average value”.

(12) Line 122. Authors should add “when the averaging time exceeds 100 s”. Indeed, if averaging time is lower than 100 s, instabilities of TSC and DISC are the same, as can be seen from Fig. 2(b).

Response: In the revised manuscript, we added “when the averaging time exceeds 100 s” in line 311.

(13) Line 167. Replace “SECM” with “SCME”.

Response: We corrected it.

(14) Line 172. Replace “possible” with “possibility”.

Response: We replaced “possible” by “possibility” in line 387 in the revised manuscript.

(15) Line 174. I find “terrible” a little bit inappropriate term. Use “harsh”.

Response: We replaced “terrible” by “harsh”.

(16) 3(a). I cannot see in the plot the experimental data referring to Pe = 0.4, which authors introduced (line 162) as the excitation fraction corresponding to the regular operation of the clock. Why? Being Δuf values @ Pe = 0.4 reported in Fig. 3(b) the corresponding average Δuf value should be also reported in Fig. 3(a) for completeness.

Response: When we obtained the data in Fig. 3(a), we did not measure the collision shift at the excitation fraction corresponding to the regular operation of the clock (Pe=0.4), considering that we will carefully measure it in the following work. The data of Fig. 3(b) can be directly applied in the Fig. 3(a). Thus, we just use the weighted average of Fig. 3(b) to supplement the correspond data in Fig. 3(a) in the revised manuscript.

(17) Conclusion is not a conclusion, but a second abstract. It should be removed in the present form, or replaced with a few considerations different from a mere repetition of the aim of the work and the obtained results. For instance, authors should add the future perspective of their research activity, if any (e.g. are there feasible ways to reduce offsets when drift rate changes rapidly?).

Response: Thank you for your comments and suggestions. We replaced the conclusion by “In summary, we propose and experimentally demonstrate a drift insensitive self-comparison method to eliminate the self-comparison measurement error. By measuring the frequency difference between two identical clock loops, the self-comparison measurement error is (240±34) mHz by the traditional way, while it is (-15±16) mHz by the drift insensitive way. Based on the drift insensitive self-comparison technique, we used Rabi spectrum to evaluate the collisional shift of our clock. By fixing the excitation fraction at about 0.4, we can determine the magnitude of the collisional shift with a precision of 2.8×10-17, even though no drift compensation is applied. The DISC method could be widely used in the evaluation of the space clocks that have less possibility to operate the clock under a condition of no drift or a linear drift due to complicated space environment. Combining with the drift compensation process, the SCME can be further suppressed even in a harsh experimental environment”.

       When the drift rate change obviously during one clock feedback cycle, it seems that no method can be used to further reduced the SCME, as one clock feedback cycle is the minimum period to update the drift rate of the drift compensation system. Thus, to avoid stray field or the clock laser frequency drifts by well controlling the experiment environments is inevitable.

We carefully checked the manuscript to avoid any grammar or spelling mistakes. All revisions have been highlighted by using the “Track Changes” function in the revised manuscript.

Reviewer 3 Report

In this paper, the authors address the issue of the drift of the clock laser between interleaved interrogation sequences used for the self evaluation of systematic effects in optical clocks.

It is a very well known and obvious fact that drifts of the clock laser frequency can cause measurement errors, especially in self-comparisons. To my knowledge, this problem is usually solved by implementing a digital double stage integrator (noted DI below), and/or a feed-forward de-drift of the clock laser. This technique basically solves the issue with very little overhead, not only for self-comparisons, but also for any use of the clock output, e.g. for comparing different clocks. Because this technique only compensates for linear drifts, quadratic drifts may yield a residual inaccuracy or instability in the feedback loop.

Here the authors propose to circumvent measurement errors due to the drift of the clock laser by a simple re-ordering of the interrogation sequence (DISC method), instead of the conventional DI. This technique has the advantage that it is a bit simpler than the DI scheme (it is conceptually simpler to visualize, and it might spare writing a couple of lines of code...). But I have several objections to this method and to the way the authors present it:

1/ I think the main problem of the paper is that the authors do not provide a quantitative comparison between the DISC method and the DI method. They say that "when the drift rate varies fast, this frequency compensation method cannot completely eliminate the SCME", but this claim is not supported by any evidence in the paper. They briefly mention that they tried to a apply "compensation method" yielding an offset of "more than 30 mHz", whose difference with the 15 mHz of the DISC method is, I think, not statistically relevant, all the more if the two methods have been applied to different data sets (this I guessed, because it is not explained in the paper)
My gut feeling is that the DISC method is formally very similar to a DI scheme, and that both methods will suffer the same limitations when it comes to quadratic drifts. I think that a fair comparison between the methods could be achieved by running the algorithms on simulated data.

2/ Implementing the DISC method solves the issue of the clock laser drift for self comparison measurements, but it does not solve this issue for any other use of the clock (e.g. clock comparisons...). For these, proper DI and/or feed-forward have to be implemented anyway. Once it is done, the DISC method becomes useless.

3/ in Jun Ye's group, they also use a technique based on a astute interleaving of interrogation sequences. It is maybe described in ref [19], or other publications of this group. It looks significantly more sophisticated than the DISC method used by the authors. I think the authors should at least comment on the difference between the DISC method and Ye's method, and possibly compare them with simulated data.

4/ A minor remark: It seems to me that the "unit fluorescence intensity" used in Fig. 3 is a technical scaling of the detection system used by the authors. It would be better to use a universal scale (e.g. per atom)

In conclusion: the authors propose and apply an interleave interrogation sequence that make self-referenced measurements immune to drift of the clock laser, but do not give quantitative evidence that it is an improvement over the ubiquitous way this issue is resolved in the optical clock community. For this reason, I do not recommend this paper for publication.

Author Response

1) I think the main problem of the paper is that the authors do not provide a quantitative comparison between the DISC method and the DI method. They say that "when the drift rate varies fast, this frequency compensation method cannot completely eliminate the SCME", but this claim is not supported by any evidence in the paper. They briefly mention that they tried to a apply "compensation method" yielding an offset of "more than 30 mHz", whose difference with the 15 mHz of the DISC method is, I think, not statistically relevant, all the more if the two methods have been applied to different data sets (this I guessed, because it is not explained in the paper)
My gut feeling is that the DISC method is formally very similar to a DI scheme, and that both methods will suffer the same limitations when it comes to quadratic drifts. I think that a fair comparison between the methods could be achieved by running the algorithms on simulated data.

Response: In fact, the residual SCME of the DI scheme was various measurement by measurement due to the irregular changed drift rate. The demonstrated result of ‘more than 30 mHz’ of the SCME of the DI method indicated the minimum observed residual SCME over seven measurements (it may be smaller if increased measurement times, but it seemed much random). However, we believed that this will not prevent us to demonstrate the validity of the DISC method, as the DISC method was much different from the DI method. The DI method was feasible when the condition of linear drift was satisfied in typically ten clock feedback cycles. In contrast, the DISC method was effective as the condition of linear drift was satisfied in one clock feedback cycle. Thus, the DISC method was expected to reduce the SCME more completely than the DI method, as during one clock feedback cycle, the approximation of linear drift was generally reasonable. Moreover, the DISC method didn’t need extra devices.

2) Implementing the DISC method solves the issue of the clock laser drift for self-comparison measurements, but it does not solve this issue for any other use of the clock (e.g. clock comparisons...). For these, proper DI and/or feed-forward have to be implemented anyway. Once it is done, the DISC method becomes useless.

Response: The DISC method, can only apply to self-comparison that was widely used in the systematic uncertainty evaluation of optical lattice clocks, and it was expected that this technique can improve the self-comparison measurement accuracy, indeed, for all optical lattice clocks. The DISC method provided another option to reduce the SCME which was expected to worker better than the DI method in the self-comparison.

3) in Jun Ye's group, they also use a technique based on an astute interleaving of interrogation sequences. It is maybe described in ref [19], or other publications of this group. It looks significantly more sophisticated than the DISC method used by the authors. I think the authors should at least comment on the difference between the DISC method and Ye's method, and possibly compare them with simulated data.

Response: The method of ‘point-string analysis’ was used to remove the SCME in reference [20]. However, this method, indeed, was a process of afterward data processing, which was essentially different from the DISC and DI methods as the ‘point-string analysis’ method required linear drift in the whole process of closed-loop operation of the clock. In this experiment, the drift changed rapidly and irregularly, indicating that the ‘point-string analysis’ cannot be used to remove the SCME.

4) A minor remark: It seems to me that the "unit fluorescence intensity" used in Fig. 3 is a technical scaling of the detection system used by the authors. It would be better to use a universal scale (e.g. per atom)

In conclusion: the authors propose and apply an interleave interrogation sequence that make self-referenced measurements immune to drift of the clock laser, but do not give quantitative evidence that it is an improvement over the ubiquitous way this issue is resolved in the optical clock community. For this reason, I do not recommend this paper for publication.

Response: The evaluation of the collisional shift is unique for each clock. Considering that it is difficult to obtain the absolutely atomic number, the "unit fluorescence intensity" seems more accurate and convenient for the collisional shift evaluation and many groups also have used similar expression [R1-R3].

We can expect that the DISC method can reduced the SCME better than any other methods benefiting of its short feedback cycle. However, we did not try to replace the DI method by the DISC method, as the DI method can reduce not only the SCME but also the servo error when the frequency drift rate changed slowly enough. But in self-comparison alone, the DISC can reduce the SCME better and the SCME will be further reduced by combining the DISC and DI method.

Reference:

[R1] Lin, Y.G., Wand, Q., Li, Y., et al. First Evaluation and Frequency Measurement of the Strontium Optical Lattice Clock at NIM. Chin. Phys. Lett. 2015, 32. 090601.

[R2] Falke, St., Schnatz, H., Vellore Winfred, J.S.R., et al. The 87Sr optical frequency standard at PTB. Metrologia, 2011, 48, 399–407.

[R3] Chang Y.P., Dai-Hyuk Y., Won-Kyu L., et al. Absolute frequency measurement of 1S0(F = 1/2)–3P0(F = 1/2) transition of 171Yb atoms in a one-dimensional optical lattice at KRISS. Metrologia, 2013, 50, 119–128.

Reviewer 4 Report

The paper deals with an improved method to correct for some deficiencies presently affecting Opt. Latt. Clock measurements. I think this observation by the authors could be decisive for future applications and should be given an opportunity to reach the community of experts and interested readers.

I summarize my comments below, where I have identified a number of English language issues which need to be corrected. In many cases the meaning is not clear due to the poor language. I have made a detailed description of each sentence, providing what I think it is a more appropriate exposition of terms. Please check carefully each point.

List of suggestions (In parenthesis the line to which they apply):

(27) An optical lattice clock is not only...
(28) its ultra-low uncertainty and instability...

[5,6,6a] --> Add a recent paper on GR prediction measurements based on
                  Opt. Latt. Clocks from Ref.[16]:
                  [6a] H E Roman, Time dilation effect... PRD 102, 084064 (2020).

(67) R1. In the same way, ...., where the clock operates...
(73) ...drift accumulates over time...
(74) of R2 is twice that of R1.

(105) ...the atoms are spin-polarized...
(110) ...and the direction is along the one of the magnetic...
(112) 2.7 uK,
(118) R1 and R2 are the same)
(119) If there are no drifts at all, the expected frequency difference...
(129) ...method is lower than the one using the TSC method.

Question: In Fig.2b, is there a simple explanation of why the TSC points
(blue squares) get flat above 100s, to fall again above 1000s? Please give som hints.

(150) clock. The clock transition benefits from the detection of thousands of
trapped cold-atoms simultaneous optical transitions, resulting in optical
(151) lattice clocks with ultra-low quantum projection noise limits.

(157-158) ...by varying the current of the Zeeman slower?
                Not a clear statement. Please correct.

(177) ...as the clock regularly operates,...
(197) ...the collisional shift is obtained with a precision of...
(200) ...Combined with the drift compensation process,..

Author Response

(27) An optical lattice clock is not only...
(28) its ultra-low uncertainty and instability...

Response: We replaced “Optical lattice clock is not only…. due to their ultra-low uncertainty and instability” by “An optical lattice clock is not only…. due to its ultra-low uncertainty and instability” (in line 27).

[5,6,6a] --> Add a recent paper on GR prediction measurements based on
                  Opt. Latt. Clocks from Ref.[16]: 
                  [6a] H E Roman, Time dilation effect... PRD 102, 084064 (2020).

Response: We added this reference to cite the latest research of the general relativity based on optical lattice clock in the revised manuscript (in line 29).

(67) R1. In the same way, ...., where the clock operates...

Response: We replaced “…closer to the resonance of the R1. By the same way, after the third…where the clock operate...” by “…closer to the resonance of the R1. In the same way, after the third…where the clock operates...” (in line 88).

(73) ...drift accumulates over time...
(74) of R2 is twice that of R1.

Response: We replaced “…domain and the amount of the frequency drift accumulated over time…of R2 is twice than R1.” by “…domain and the amount of the frequency drift accumulates over time…of R2 is twice that of R1.” (in lines 94-95).

(105) ...the atoms are spin-polarized...

Response: We replaced “...the atoms is spin-polarized...” by “...the atoms are spin-polarized...” (in line 136).

(110) ...and the direction is along the one of the magnetic...
(112) 2.7 uK, 

Response: We replaced “…and the lattice laser are linear and its direction is along the direction of the magnetic…the longitudinal and radial temperatures of atoms trapped in the lattice are 2.9 μK and 2.7 μK respectively.” By “…and the lattice laser is linear and the direction is along the one of the magnetic…the longitudinal and radial temperatures of atoms trapped in the lattice are 2.9 μK and 2.7 μK respectively.” (in line 141).

(118) R1 and R2 are the same)
(119) If there are no drifts at all, the expected frequency difference...

Response: We replaced “…between the two identical clock loops (the systematic parameters of R1 and R2 is same)…If there are not any drifts, the expected frequency different…” by “…between the two identical clock loops (the systematic parameters of R1 and R2 are the same)…If there are no drifts at all, the expected frequency different…” (in lines 149-150).

(129) ...method is lower than the one using the TSC method.

Question: In Fig.2b, is there a simple explanation of why the TSC points

(blue squares) get flat above 100s, to fall again above 1000s? Please give some hints.

Response: We replaced “…method. So, the self-comparison instability of the DISC method will lower than the TSC method.” By “…method. So, the self-comparison instability of the DISC method is lower than the one using the TSC method.” (in line 173).

       From the inset of the Fig. 2(a), we can find that the clock laser frequency drift rate changed rapidly and irregularly when the clock initially operated, but the clock laser frequency drift seemed much linear in the whole closed-loop operation. And the frequency drift rate determined the frequency offset (the data used in Fig. 2(b)) of the TSC method. Thus, during 1~100 s, the self-comparison instability of the TSC averaging down with the increasing of the measurement time, as the short-term instability, mainly determined by the clock laser, was less insensitive to the drift. Then, the changed drift rate caused that the instability became flat during 100~1000 s. The long-term instability fallen again above 1000 s due to the approximately linear drift in the whole process, because the frequency offset difference between data set (each of which was the average over more than 1000 s) was reduced compared with the case of 100~1000 s. The spinodal depended on how did the clock laser frequency change, and thus, it could be different between measurements.

       In the revised manuscript, we added “The instability falls again above 103 s due to the approximately linear drift in the whole process.” in the caption of the Fig. 2(b) (in lines 191-192) to explain it.

(150) clock. The clock transition benefits from the detection of thousands of 
trapped cold-atoms simultaneous optical transitions, resulting in optical
(151) lattice clocks with ultra-low quantum projection noise limits.

Response: We replaced “…clock. benefiting of detecting the clock transition of thousands of trapped cold-atoms simultaneously, the optical lattice clocks have ultra-low quantum projection noise limits.” by “…clock. The clock transition benefits from the detection of thousands of trapped cold-atoms simultaneous optical transitions, resulting in optical lattice clocks with ultra-low quantum projection noise limit.” (in lines 195-196).

(157-158) ...by varying the current of the Zeeman slower? 
                Not a clear statement. Please correct.

Response: We replaced “...by varying the current of the Zeeman slower.” by “...by varying the currents of the Zeeman slower, which changes the loaded atoms of the first-stage cooling and eventually changes the total atomic number of the lattice” (in line 203).

(177) ...as the clock regularly operates,...

Response: We replaced “…as the clock regular operates,…” by “…as the clock regularly operates,…” (in line 231).

(197) ...the collisional shift is obtained with a precision of...

Response: We replaced “…the collisional shift can be determined with a precision of…” by “…the collisional shift is obtained with a precision of…” (in line 234).

(200) ...Combined with the drift compensation process, …

Response: We replaced “…Combining with the drift compensation process, …” by “…Combined with the drift compensation process, …” (in line 263).

Round 2

Reviewer 1 Report

Thank you for your answer. I think still there is misunderstanding between us. The manuscript presents only one systematic shift which is not described as a first. I see that other groups that work with optical clocks present in one publication all effects that are composed in one accuracy budget. In my opinion the scientific content of the manuscript must be still improved a lot. If you publish only one effect I expect that an evaluation of this effect should be very deep. Moreover methods should be clearly described. The same problem is with the DISC method. Please remember that you publish your research not for you but it is a form to communicate with the world. I would like to learn something new by reading your publication and some people want also to compare own results with yours because we believe that atomic properties are the same in the whole Universe and effects in optical clocks should be comparable. This is one of reasons why we use theoretical models: models applied in an independent machine should work as well. Otherwise your results cannot be independently verified and are useless for the community. Here I see no readiness to raise the quality of the publication. I recommend to reject this article.

Author Response

This paper demonstrated a drift insensitive self-comparison method that had been detailedly described in the main text. Based on that the experimental data had shown the validity of this method, we applied this method to measure the collisional shift in our clock to further experimentally verify this method. If the residual SCME was obvious, large difference between measurements was expected as the clock laser frequency drift rate was different between measurements. The theory of the collisional shift, which had been widely studied [30,32,33,R1,R2,R5], can help us to phenomenological understand this shift in optical lattice clocks, however, in this paper, we focused on the DISC method and the experimental measurement of the collisional shift by the DISC method. We emphasized that, the DISC method can be widely used in all optical lattice clocks and especially in the transportable clocks and the space clocks. It is worth to note that the collisional shift approximately linearly depends on the excitation fraction [R1] (in the third paragraph). And there were many groups that use this linear approximation to deal with the relationship between the excitation fraction and the collisional shift [R2-R4]. In terms of the collisional shift, the measured results of different groups were lack of comparability even for the same element (such as 87Sr), because they used different lattice structure, temperature, atomic density, population purity (spin-polarized process), excitation fraction and so on. The measured s-wave or p-wave scattering length (such as for 87Sr) even different measured by different groups or setups [R2,R5]. This was why every group will evaluate the collisional shift of their clock, but few groups compared their result with else groups [4,17,20,21,22,24]. Reference: [R1] Ludlow, A.D., Lemke, N.D., Sherman, J.A and Oates, C.W., Cold-collision-shift cancellation and inelastic scattering in a Yb optical lattice clock, Phys. Rev. A. 2011, 84, 052724. [R2] Zhang, X., Bishof, M., Bromley, S.L. Spectroscopic observation of SU(N)-symmetric interactions in Sr orbital magnetism. Science, 2014, 345, 1467. [R3] Rey, A.M., Gorshkov, A.V., et al. Probing many-body interactions in an optical lattice clock. Ann. Phys. 2014, 340, 311–351. [R4] Gao, Qi., Zhou, Min., Han, C.Y., et al. Systematic evaluation of a 171Yb optical clock by synchronous comparison between two lattice systems. 2018, Sci. Rep. 8:8022. [R5] Rey, A.M., Gorshkov, A.V., and Rubbo, C. Many-Body Treatment of the Collisional Frequency Shift in Fermionic Atoms. Phys. Rev. Lett. 2009, 103, 260402.

Round 3

Reviewer 1 Report

Thank you for your answer. Groups that focus only on an evaluation of the collisional shift show also a theoretical model. On the other hand groups that show only experimental results show also other systematic effects that are composed in one accuracy budget. Your manuscript is none of the above. I repeat: in my opinion the scientific content of the manuscript must be still improved a lot. Moreover there is no scientific reason to present the DISC method for an evaluation of the collisional shift because the DISC method can be applied to any systematic shift in an optical clock. I highly recommend to present a total accuracy budget of your clock with the DISC method. This would be a great publication. Good luck.